# Overcome low levels of detection limit and choice of antibody affects detection of lipoarabinomannan in pediatric tuberculosis

Anita G. Amin[1]⊕, Prithwiraj De[1]⊕, Barbara Graham[1], Brooke L. Jensen[1], Emmanuel Moreau[2], Delphi Chatterjee[1]*

1 Department of Microbiology, Immunology and Pathology, Mycobacteria Research Laboratory, Colorado State University, Fort Collins, Colorado, United States of America, 2 Foundation of Innovative New Diagnostics, Geneva, Switzerland

⊕ These authors contributed equally to this work.
* delphi.chatterjee@colostate.edu

**Data Availability Statement:** All relevant data are within the manuscript and its Supporting information files.

## Abstract

The World Health Organization (WHO) emphasizes that tuberculosis (TB) in children and adolescents is often overlooked by healthcare providers and difficult to diagnose. As childhood TB cases rise, finding a diagnostic high in sensitivity and specificity is critical. In this study 91 urine samples from children aged 1–10 years were analyzed for tuberculostearic acid (TBSA) by gas chromatography/mass spectrometry (GC/MS) and capture ELISA (C-ELISA). In C-ELISA the CS35/A194-01 antibody performed very poorly with both curve-based and model-based cutoffs. The area under the ROC curve (AUC) of the CS35 $OD_{450}$ values was only 0.60. Replacing the capture antibody with BJ76 gave a better performance in both sensitivity and specificity (AUC = 0.95). When these samples were analyzed by GC/MS, 41 classified as 'probable/possible' for TB were distinctly TBSA positive with ten samples having <3 ng/mL LAM. However, from the 50 samples with 'unlikely' TB classification, 36 were negative but 7 had >3 ng/mL and were designated as LAM positive. This experimental assay assessment study signifies that i) the antibody pair CS35/A194-01 that has been successful for adult active TB diagnosis is not adequate when LAM level is low as in pediatric TB; ii) no one mAb appears to recognize all TB-specific LAM epitopes.

## Introduction

The SARS-COV-2 (COVID-19) pandemic impacted health and economics in 2020 and 2021, including access to essential tuberculosis (TB) services such as national disease surveillance systems and reports on TB disease burden (incidence and mortality). The World Health Organization (WHO) *Global tuberculosis report 2020* [1] included provisional estimates of TB incidence in 2020 suggesting that a global total of about 10 million people fell ill with TB in 2020. Children also fall victim to this disease as evidenced by the 1.2 million pediatric cases seen worldwide in the same year. TB in children and adolescents is often overlooked by healthcare

**Funding:** This work was funded through NIH AI R01 AI132680 (to DC) and the samples analyzed in this study were provided under NIH/NAID U19 AI109755 to Dr. Molly F Franke (Harvard Medical School). The funders had no role in study design, data collection and analysis, decision to publish, or preparation of the manuscript.

**Competing interests:** The authors have declared that no competing interests exist.

providers and difficult to diagnose [2]. The greatest numbers of pediatric TB cases are seen in children younger than 5 years, and in adolescents older than 10 years.

Childhood TB differs from adult TB due to its differential signs, which are based on the child's age [3]. High risk factors for contracting latent TB include contact with an adult who is likely infected or exposed to TB, or traveled to a country where TB is endemic [4]. Healthcare providers needs to consider the clinical signs such as fatigue, weight loss, lack of playfulness, persistent cough, and a positive tuberculin skin test (TST) [5]. A screening questionnaire serves as one of the first steps in determining a child's risk for TB and latent TB [4]. If the questionnaire indicates one or more risk factors, TST is performed followed by a medical examination and medical history; then chest radiographs to confirm the diagnosis of TB [4].

Immunodiagnostic assays are another means to diagnose childhood TB [4]. The Microscopic Observation Drug Susceptibility (MODS) assay, for instance, is an immunodiagnostic test that uses either gastric fluid aspirate or sputum samples from pediatric patients [6]. However, a major drawback to MODS is that the required sputum samples are difficult to obtain from young children [6].

Two commercial blood based Interferon-gamma-release-assays (IGRA) in use are the T-SPOT and the QuantiFERON-TB Gold In-tube test [7]. When both an IGRA and TST were administered to children aged 5 to 18, the sensitivity was higher (96%) than for TST alone (83%) [8].

Recently, urine samples have been shown to be useful in the diagnosis of pediatric TB. A study performed by Gautam *et al.* used the Alere Determine™ (AlereLAM) assay to detect LAM in urine from children infected with intrathoracic TB (ITTB) or lymph node TB (LNTB) [9]. AlereLAM detected 21 LNTB and 46 ITTB cases that were missed by other methods including Ziehl-Neelsen (ZN) stain, liquid culture, and Xpert testing.

A second immunodiagnostic test to detect the presence of LAM in urine is the Fujifilm SIL-VAMP TB LAM test (FujiLAM). Nicol *et al.* looked at the sensitivity and specificity of FujiLAM in children suspected of having a TB [10]. The sensitivity of FujiLAM (42%) and AlereLAM (50%) were similar [10]. However, FujiLAM had higher sensitivity in children with HIV and children who were malnourished. FujiLAM additionally had significantly higher specificity (92%) compared to AlereLAM (66%). Because of its high specificity, it has been suggested that the FujiLAM test may be useful to "rule-in" children in high-risk situations including those who are malnourished or living with HIV [10]. With childhood TB cases on the rise, finding a diagnostic approach that is high in both sensitivity and specificity is critical.

In our continued effort to validate LAM as a biomarker for active TB, in this study we examined 91 urine samples from children (age range 1–10 yrs). Clinical diagnosis was made by a pediatric pulmonologist based on smear, culture, chest-X-ray, TST, contact history, and symptoms. All samples were analyzed simultaneously by gas chromatography/mass spectrometry (GC/MS) and C-ELISA, the two assays that have been utilized and applied over time in our laboratory [11–13]. All urine samples were from sputum smear negative children. Our goal was to see if LAM can be quantified in the pediatric urine samples and if the C-ELISA could be applied successfully to detect LAM.

## Methods and materials

### Sample cohort

Anonymized urine samples used in our study were provided by Laboratorio Socios En Salud Sucursal, Peru. The study samples were collected from children with suspected TB and symptomatic children in whom TB was ruled out by a pediatric pulmonologist on the basis of negative bacteriological (*i.e.*, smear and culture) results from sputum or gastric aspirate; chest x-

ray, and tuberculin skin testing (n = 91). All samples were collected in the morning. Sample size was not predetermined, we received and analyzed what was available as left over and sent to CSU for our experiments [14].

Urine for LAM spiked and negative control samples was obtained from healthy volunteers from a TB non-endemic region, centrifuged within an hour of collection and the supernatant aliquoted and stored frozen at -80˚C until further use. To derive a LAM assay standard curve, the control urine (NEU) was spiked with CDC1551 LAM that is serially diluted two-fold to obtain a concentration range (range 0.02–12.5 ng/mL). The same urine sample was used as a negative control (no spike) for the urine background to derive a cutoff for the analysis of clinical samples. A standard curve was routinely extrapolated every time a new antibody was tested with each set of clinical samples.

## Ethics statement

The study generating the urine samples conformed to the Declaration of Helsinki and was approved by the Ethics Committee of the Peru National Institute of Health and the Institutional Review Board of Harvard Medical School. Written informed consent was provided by adults and guardians. The present research was approved by the CSU Institutional Biosafety Committee (IBC), and the CSU Integrity and Compliance Review Board (IRB) under approval IRB protocol number 09-006B.

## LAM for assay standardization

The LAM used in this study was isolated and purified from *Mycobacterium tuberculosis* (*Mtb*) CDC1551 in large quantities (20–30 mgs of LAM isolated from ~90g of wet cells) so that the same standard was used throughout the year for recurring experiments [15].

## Pretreatment of urine samples

Sample aliquots were thawed on ice before use. A 100μL aliquot of the urine sample was transferred to a fresh screw-cap tube and dialyzed, using a 3K MWCO dialysis tubing (Spectraphor), against DI $H_2O$ overnight followed by treatment with Proteinase K (Pro K, Thermo Fisher Scientific) at a final concentration of 200μg/mL at 55˚C for 30 min and then inactivated by boiling (100˚C) for 10 min. The supernatant obtained after centrifugation at 12,000 x g for 10 min was directly used for ELISA. The control samples were analyzed in triplicate and the clinical samples were analyzed in duplicate. For the adult clinical samples, the dialysis step was omitted and the ProK pretreatment was carried out before the C-ELISA analysis.

## Antibodies (mAb)

Our experiments used a previously described mouse mAb CS35 IgG3 raised against *Mycobacterium leprae* was obtained from the CSU repository [11, 16], subcloned and purified by Pinter laboratory (Rutgers University) and used for our experiments. A human mAb A194-01 IgG1 was obtained from Pinter laboratory (Rutgers University) [17]. Application and use of the antibodies has been described in our previously published works [11, 12, 18]. A rabbit monoclonal antibody, BJ76 IgG was obtained from Foundation for Innovative New Diagnostics and as reported, and was raised by immunizing rabbits with the cell wall components of *Mycobacterium tuberculosis* $H_{37}Rv$ [19]. A single chain fragment variable (scFv) phage display library was generated, the BJ76 nucleotide sequence was deducted from selected clones from the library [19] and cloned into an expression vector for production and purification of a rabbit IgG/k in CHO cells. No animal work was undertaken for generation of these antibodies.

## Capture ELISA (C-ELISA)

A 96-well polystyrene high binding microplate (Corning, costar) was coated with the capture antibody (CS35 at 10µg/mL and BJ76 at 2.5µg/mL) overnight at 4˚C. The plates were blocked with 1% BSA for 60 min after a brief rinse with the same. Urine spiked with LAM and the clinical samples (pretreated) were applied to the appropriate wells on the plate/s and incubated at 27˚C for 90 min. After incubation, the plate/s were washed (1X PBS, Tween 80) and detection antibody (biotinylated A194-01 @ 250 ng/mL) was applied and incubated for 90 min at 27˚C. The incubation was followed by washes and a 25 min incubation with streptavidin horseradish peroxidase (R&D Systems) @ 27˚C. Ultra TMB-ELISA, a chromogenic substrate (Thermo Scientific) was added to the plate/s to develop the color for 30 min and the reaction was stopped by adding sulphuric acid and the optical density measured at 450nm (for the sample) and 570nm (background signal from the plate).

## Gas chromatography/mass spectrometry analysis for TBSA in urine LAM

The method developed has been described in detail [13, 20]. We modified the method for this particular set of samples by introducing a hexane extraction step prior to the hydrophobic interaction chromatography (HIC). Typically, urine samples (1 mL) after the hexane (1mL) wash were dried under vacuum and then subjected to HIC over Octyl Sepharose® (OS)-CL 4B using 5–65% step gradient of n-propanol in 0.1 M ammonium acetate ($NH_4OAc$). The 40% and 65% n-propanol in 0.1 M $NH_4OAc$ eluents from the HIC column was processed for GC/MS analysis to make the corresponding pentafluorobenzyl tuberculostearate derivative. $D_2$-palmitic acid (5ng) was used as an internal standard. The Octyl Sepharose® purified LAM from urine was dried ($N_2$-stream) and subjected to alkaline hydrolysis in 0.25 N aqueous NaOH (200 µL) and heated at 80˚C for 1.5 h. Water (0.5 mL) was added to the tube and acidified with conc. $H_2SO_4$ (5 µL) to pH ~2. It was then extracted with $CHCl_3$ (500 µLx2) and organic layers were collected together and dried. To neutralize the trace amount of acid, 28% $NH_4OH$ (100 µL) was added to it and dried again. The dry sample was then treated with acetonitrile (100 µL), diisopropyl ethylamine (50 µL) and 2,3,4,5,6-pentafluorobenzyl bromide (20 µL) at room temperature. After 30 min, the sample was dried ($N_2$-stream) and reconstituted in Hexanes (AR Grade) (100 µL) and then transferred to GC vials for GC/MS.

The GC/MS analysis of the pentafluorobenzoate ester was carried out using selective ion monitoring program in a negative chemical ionization-splitless mode. The inlet temperature was at 250˚C. The oven was set at 50˚C for 1 min, ramped at 20˚C/min till 150˚C and held for 0.01 min and finally ramped at 10˚C/min to 310˚C and held for 5 min. Methane was used as the collision gas at 2.0 mL/min. The source and MS transfer line temperatures were 250˚C and 330˚C respectively. The instrument was set to collect data for $m/z$ 257.3 (internal standard) in the range of 5 to 18.5 mins and $m/z$ 297.3 from 18.5 to 27.1 mins for TBSA. GC/MS chromatograms with respective peaks were integrated manually and integrated areas were generated by the Chromeleon v 7.2 software for the estimation of total TBSA content. It was then used to calculate the LAM-equivalent using previously reported formulae [13].

## Statistical analysis

BJ76 was tested twice and the $OD_{450}$ values averaged. One sample had insufficient sample for a second run and only the first run value was used. Cutoffs for classification using $OD_{450}$ and TBSA measurements were determined in three ways: from standard curves of LAM-spiked urine; from empirical data to maximize area under a receiver operating characteristic (ROC) curve using the ROCR package [21] version 1.0.11, and calculated as inflection points in logistic regression models using base R. ROCR was also used to calculate 10-fold cross-validation

ROC curves. The standard curve cutoff for BJ76 was an average of cutoffs for the two runs. Confidence intervals for sensitivity and specificity were calculated using Wilson's method [22] with a continuity correction, as implemented in the DescTools package [23] version 0.99–43. ROCR and the ggplot2 package [24] version 3.3.5 were used to create plots. Statistical analyses were conducted in R open source software version 4.1.1 [25].

# Results and discussion

## Sample cohort

The 91 pediatric TB cases had an age range of 1–10 years and were not tested for HIV. All samples were received blind and clinical status was revealed after analyses were completed. Clinical diagnoses were made by a local pediatric pulmonologist based on smear, culture, chest-X-ray, TST, contact history, and symptoms. All children samples were smear negative as reported and received frozen.

## LAM quantification of pediatric urine samples using GC/MS

D-arabinose (55–60%), D-mannose (36–40%), and fatty acyls (1–3%; palmitate C:16; tuberculostearate (TBSA) C:19:1) [26, 27] are monomeric components amenable to GC/MS after derivatization of LAM. For quantification of LAM by GC/MS in urine, we have been routinely using D-ara and TBSA as structural surrogates for the "full-length" LAM [28]. The use of hydrophobic interaction chromatography (HIC) is an integral part of our GC/MS protocol where LAM is purified away from other interfering urinary components. The criteria for calling a sample LAM positive in the GC/MS analysis of TBSA and D-ara is that the same fraction eluting off the Octyl Sepharose CL 4B® column (40–65%), must have both TBSA and D-ara in amounts that calculate to be comparable to LAM (*i.e.* molar ratio of D-ara vs. TBSA). However, for the pediatric samples reported herein, LAM concentrations were too low ($< 2$ng/mL) to estimate D-ara, as this required four ion peaks be integrated. Therefore, we relied on the uniformity of independent TBSA analyses done at two separate times, thus accounting for inter-assay variability to strengthen reproducibility and support our conclusions. TBSA assay reproducibility gives us confidence in LAM amounts present in each sample. Correlation of the two assays was excellent (Fig 1). The amount of LAM-equivalent was calculated using the previously reported formulae [13]. Based on these calculations, a majority of the pediatric TB-positive samples in this study cohort had 2–8 ng/mL of LAM in urine.

From the 91 urine samples analyzed by GC/MS, 41 that were classified as 'probable/possible' for TB were also TBSA positive with ten samples having less than 3 ng/mL of LAM (S1 Table). However, from the 50 TB negative or with 'unlikely' TB classification, 36 were clearly negative (i.e. on the mass spec the peak for TBSA at $m/z$ 297.3 @ retention time 19.5 mins- flat lined) but 7 had $>3$ ng/mL and were designated as LAM positive (S1-S13 Figs in S1 File). This observation is consistent with our published work in adults, in which we find urine samples classified as negative by clinicians to have LAM detected by GC/MS [13, 20].

## Detection of LAM in pediatric urine samples using ELISA

For these pediatric samples, we first tested the antibody pair CS35/A194-01 in C-ELISA. CS35 performed very poorly as $OD_{450}$ values were not clearly distinct between the TB groups. The majority of the samples had values near the curve-based cutoff. The area under the curve (AUC) of the CS35 $OD_{450}$ value was only 0.60.

Following this, we changed two parameters. In the second set of experiments, urine samples were first dialyzed in water, and then treated with proteinase K as described [11] and a

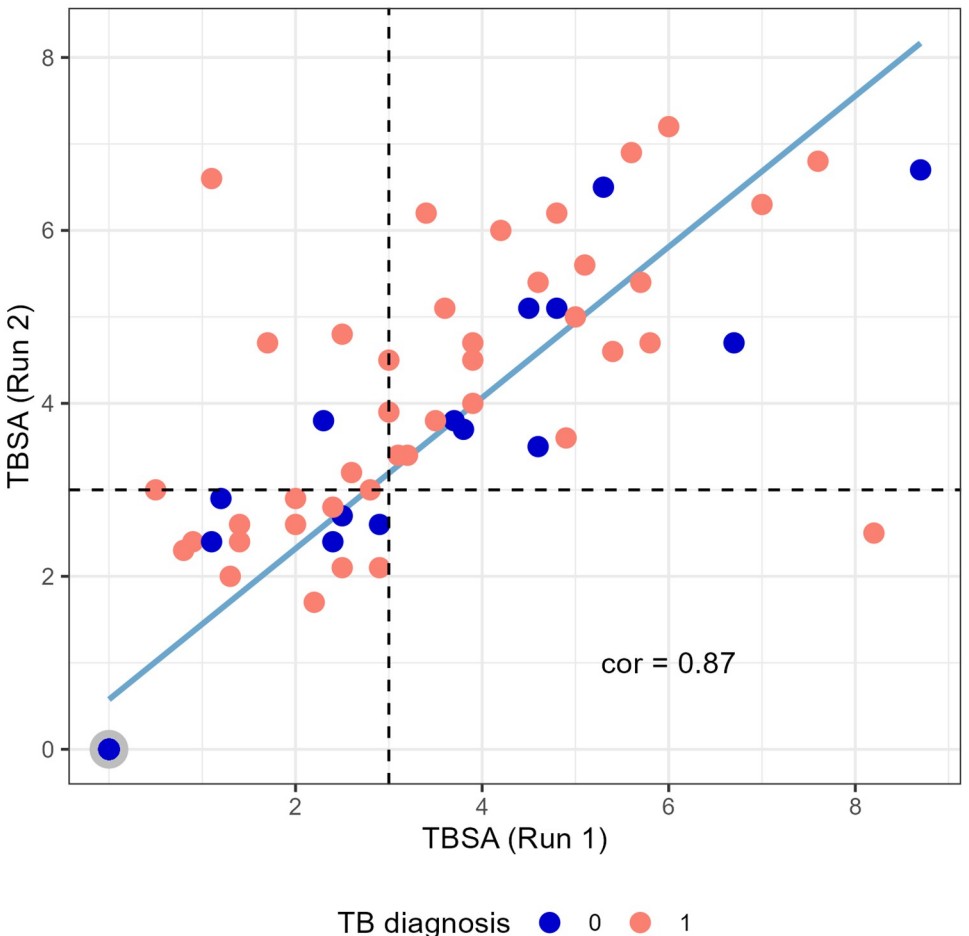

**Fig 1. Repeat TBSA analyses at two different times.** The two times TBSA assay values (ng/mL) are plotted against each other with 3ng/mL indicated by dashed lines. The Pearson's correlation validates the assay accuracy. A number of samples diagnosed as negative by clinicians (blue dots) had high levels of TBSA and therefore concluded to be LAM positive.

different capture antibody (BJ76) was used. This antibody pair was selected from our repertoire based on its comparable performance with CS35 and a good limit of detection (LoD) above the background at 25–50 pg/mL of LAM spiked in urine (Fig 2). BJ76 is a rabbit mAb provided to us by FIND and has been shown to have affinity to the terminal methylthioxylose in LAM in a glycan array (manuscript in preparation). In contrast to our published studies showing that CS35 when paired with A194-01 worked well with adult TB urine samples with an LoD of ~100pg/mL but gives high background with both unspiked urine and the clinical samples. BJ76 had significantly lower background and an LoD of ~25–50 pg/mL when used on spiked urine samples. For the clinical samples, both sensitivity and specificity of the C-ELISA improved with this pair of mAbs, having an AUC of 0.96. BJ76 performed better than CS35 in all pediatric samples (Fig 3). A comparison of the sensitivity and specificity for the two antibody pairs is shown in Fig 4.

Cutoff values were calculated by the different methods and are given for each antibody (Table 1). Balanced accuracy (the average of sensitivity and specificity) is given for the cutoffs in each model. The high cutoff value identified by the logistic model for CS35 is indicative of

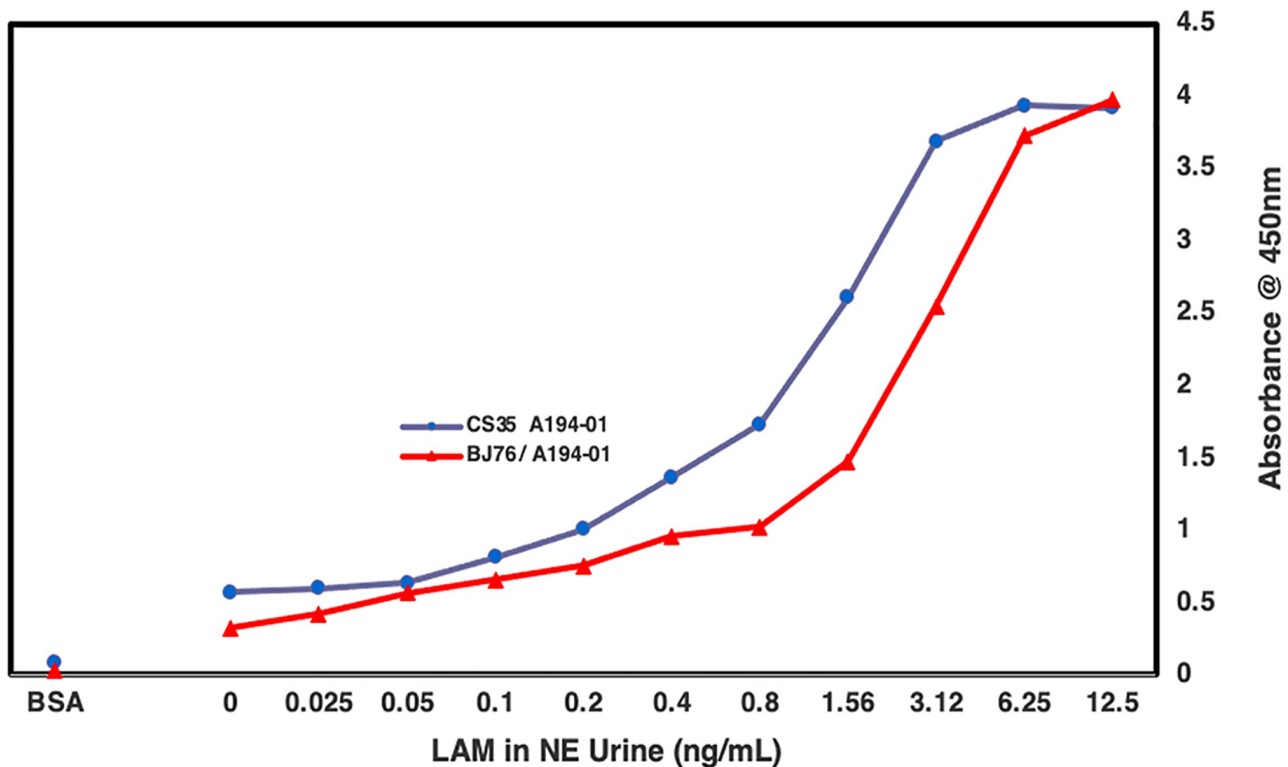

**Fig 2. Comparative ELISA OD$_{450}$ for two mAbs used for preselection in spiked urine samples.** Generation of LAM standard curve with *Mtb* CDC1551 LAM spiked into urine from a healthy volunteer using C-ELISA showing CS35, mouse mAb @ 10μg/mL as capture antibody against biotinylated A194-01, human mAb @ 250ng/mL used as the detection antibody (blue circles) in comparison to BJ76, rabbit mAb @ 2.5μg/mL as capture against biotinylated A194-01 @ 250ng/mL as the detection antibody (red triangles). The graph shows BJ76 when paired with A194-01 antibody shows reduction in background signal and improved LoD (from 100pg/mL for CS35/A194-01 pair) to about 50-20pg/mL.

the poor performance of that antibody, as OD$_{450}$ values were similar in both TB-positive and TB-negative samples.

From the TBSA quantification, we found the amounts of LAM in the pediatric population with TB disease is lower than we see in adults with TB, a finding consistent with the higher frequency of paucibacillary TB in children. The lower levels of LAM available in the urine samples of children along with the higher background signals from CS35 may be the reason for the unsatisfactory performance of the CS35/A194-01 pair as compared to BJ76/A194-01. This was further substantiated when we tested 25 adult urine samples obtained from FIND (S2 Table). These urine samples had been stored in the CSU storage since 2015. Thirteen out of 25 were culture positive with 2+, 3+ smear gradation and 12 were culture negative. With CS35/A194-01 both specificity and sensitivity were 100% correctly identifying 13 /25 as C-ELISA positive and 12/25 as C-ELISA negative whereas with BJ76/A194-01 the sensitivity dropped to 61.5% as for 5/13 culture positive samples OD$_{450}$ fell below the ELISA cut off. This supports our report that CS35 as a capture mAb is very effective in detecting LAM in adult TB patients where LAM amounts are comparatively higher (> 8ng/mL) in majority of the clinical samples. A reason for this antibody selectivity is unclear at the present time since BJ76 has not gone through a rigorous validation.

The current study signifies that i) antibody pair CS35/A194-01 that has been so successful for adult active TB diagnosis is not adequate when the LAM amounts are very low (as in

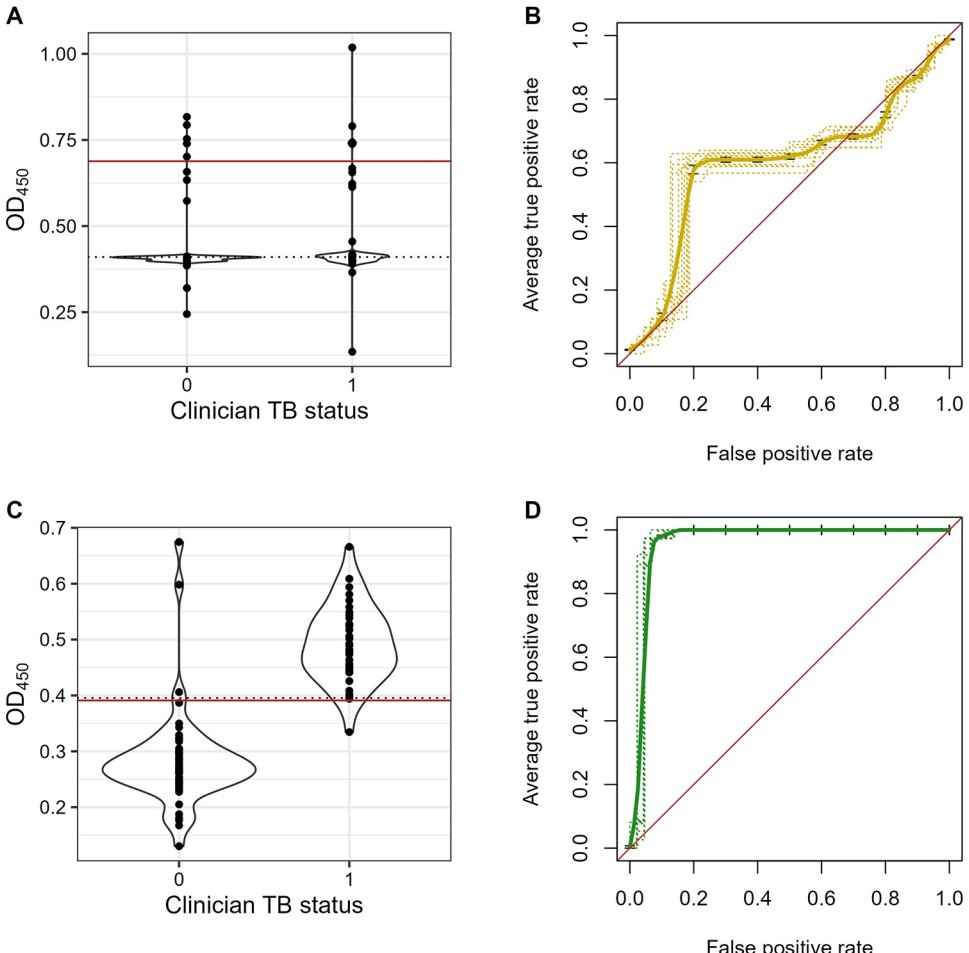

**Fig 3. Comparative performance analyses of CS35 vs, BJ76 as capture mAbs in C-ELISA. A:** values for mAb CS35 are plotted for each TB status; violin plots indicate the density of points. The cutoff based on a logistic model is shown by the solid red line; the cutoff based on the standard curve is given by the dotted line. The majority of values were at or near the standard curve cutoff. **B:** Average ROC curve with 10-fold cross-validation curves shown by dotted lines. AUC was low in all folds. **C:** $OD_{450}$ values for BJ76 are plotted for each TB status; violin plots indicate the density of points. The cutoff based on a logistic model is shown by the solid red line; the cutoffs based on the standard curve are shown by the dotted lines. **D:** Average ROC curve with 10-fold cross-validation curves shown by dotted lines. AUC was high in all folds.

pediatric TB and miliary TB); ii) sample pretreatment is not efficient in reducing background in pediatric samples giving a higher background; iii) changing the capture antibody to BJ76, seemed to have improved ELISA sensitivity for pediatric samples; iv) **no one monoclonal mAb** (to date) appears to recognize all TB-specific LAM epitopes; and v) heterogenic variation of LAM occurs naturally in urine, so that epitopes present will impact results based on mAbs in C- ELISA formats and in the LAM detection methods.

A summary of analyses of the pediatric samples is presented in the schematic flow (Fig 5). The characteristics of the 91 children included in the analysis are detailed in the S1 Table which also includes data from GC/MS (2XTBSA) and C-ELISA with two mAb pairs. The data from the 25 adult TB samples is presented in S2 Table.

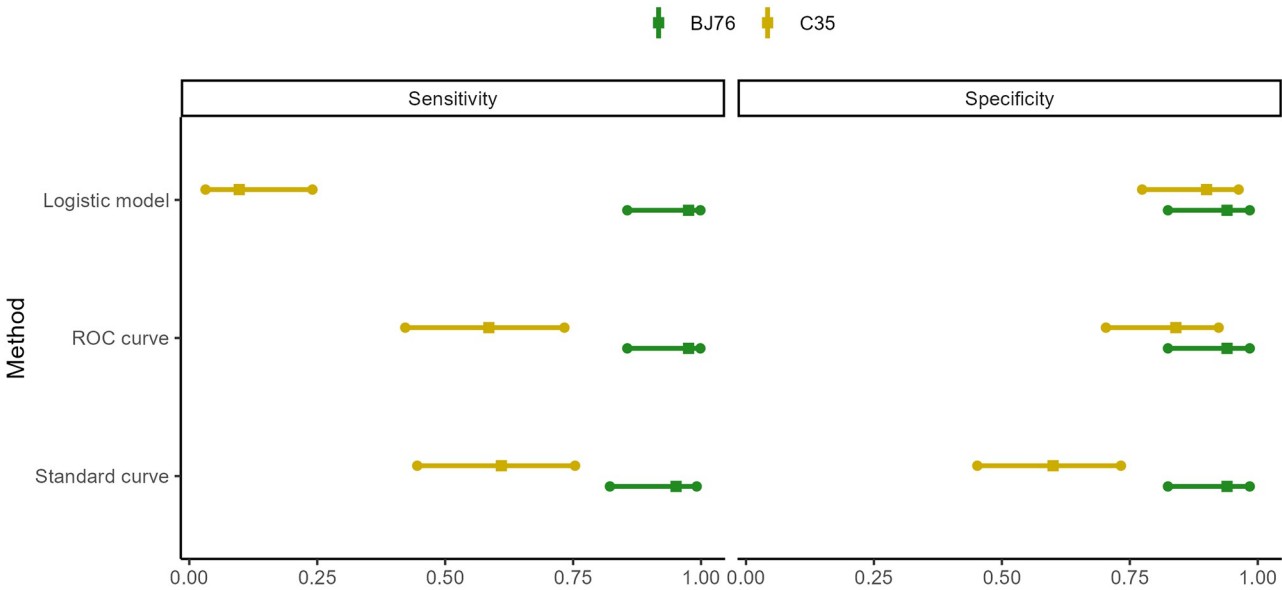

**Fig 4. A forest plot comparing sensitivity and specificity for the different models.** Sensitivity (left) and specificity (right) are plotted with 95% confidence intervals. Classifications using the standard curve cutoffs for BJ76 were made using the average of cutoffs for each run. BJ76 sets were run in duplicate and $OD_{450}$ values were averaged (original individual values are in Table 1). BJ76 performs better than CS35 in all cases.

**Table 1. Cutoff values calculated by the different methods are given for each antibody.** Balanced accuracy (the average of sensitivity and specificity) is given for the cutoffs in each model.

| | Cutoff values | | Balanced Accuracy | |
|---|---|---|---|---|
| **Method** | **BJ76** | **CS35** | **BJ76** | **CS35** |
| Logistic model | 0.391 | 0.688 | 0.96 | 0.50 |
| Standard curve | 0.396 | 0.410 | 0.95 | 0.60 |
| ROC curve | 0.394 | 0.412 | 0.96 | 0.71 |

## Conclusion

Pediatric tuberculosis (PTB) remains a major cause of morbidity and mortality globally, particularly in developing countries. Diagnosis of pediatric TB brings one of the biggest challenges to conclusive decisions on whether or not to finalize a treatment downstream. There is a substantial research gap in developing new diagnostics for children, furthering the diagnostic conundrum in resource-limited settings. PTB (*i.e.* children under 15 years of age) is a public concern as it could be a marker for recent transmission. Infants and young children are more likely to develop life-threatening forms of TB disease (*e.g.*, disseminated TB, TB meningitis). Usual diagnostic approaches are based on clinical presentations, radiographic abnormalities, contact history, and tuberculin skin test, all of which are of low specificity [29].

In the present study on our path to develop a non-sputum-based diagnostic assay, on a set of pediatric urine samples, we first used GC/MS for quantification of LAM and showed unambiguously that it is present albeit at low levels. Concomitantly, using a simple ELISA but with a specific pair of mAb BJ76/A194-01 we were able to detect LAM clearly distinguishing TB positive from 'unlikely' TB. Our work indicates that for pediatric TB diagnosis, an ELISA based assay (which uses larger sample volumes) may be more appropriate where sample concentration can be applied and thus may not be ideal in the field. It is rather puzzling why a broad

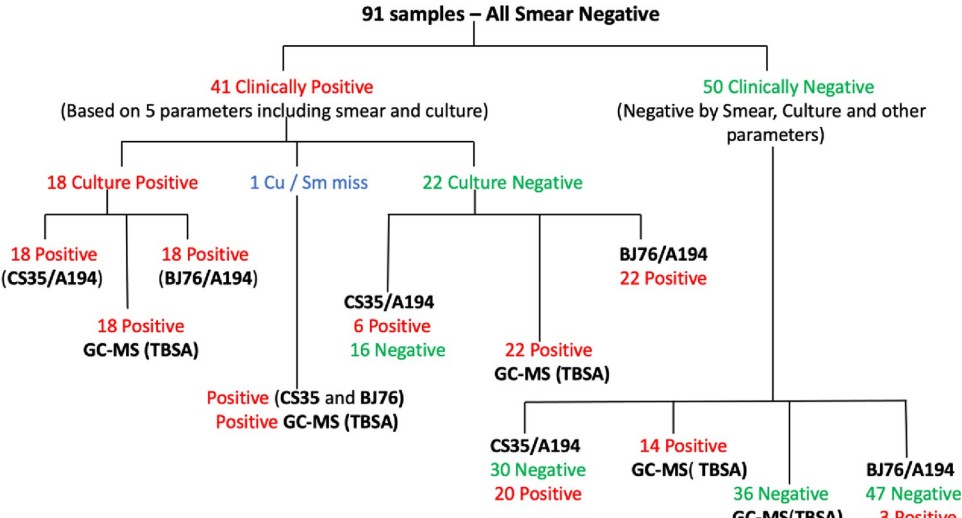

**Fig 5. Summary of analysis of pediatric TB samples in this study.** Flow chart for 91 pediatric TB samples showing clinically positive and negative status assigned by the clinicians with the patient samples diagnostic outcomes using C-ELISA and GC-MS to detect LAM in urine.

spectrum antibody like CS35 would selectively recognize LAM from the adult TB population but bind less efficiently to PTB, it is tempting to suggest that there is epitope variability, an issue that cannot be addressed easily due to low abundance of LAM in PTB. Both BJ76 and A194 are claimed to be MTX specific. How they are working in a pair is also not clear? We speculate Ab specificities to multiple LAM epitopes. Also, LAM was equally distributed across all ages by C-ELISA using BJ76 (S2 File).

The use of an antibody independent approach (*i.e.* quantification based on mass spectrometry) was vital in this work as we could assess the levels of LAM that needed to be detected in an immunoassay and thus selected an mAb from our repertoire that offered least background. Limitations of our study are that our sample cohort was from a HIV negative geographical region and that contribution of HIV could not be assessed in this particular set. A larger cohort is needed to support our conclusion. The novel assays require further evaluation in distinct clinical settings and in immunosuppressed patient groups.

## Supporting information

**S1 Data. Table of contents.**
(DOCX)

**S1 File. S1-S13 Figs.** GC/MS chromatograms of TBSA analysis of 91 pediatric urine samples includes both culture positive and culture negative samples.
(PPTX)

**S2 File. Forest plot- age vs LAM distribution using BJ76.**
(PDF)

**S1 Table. GC/MS data and ELISA OD$_{450}$ values, clinical status on all 91 pediatric urine samples.**
(XLSX)

**S2 Table. ELISA OD$_{450}$ values and clinical status for 25 adult urine samples.**
(XLSX)

## Acknowledgments

We thank Dr. Molly Franke at Harvard Medical School for providing us with the pediatric samples used for analysis. The authors wish to thank the Colorado State University Analytical Resources Core (RRID: SCR_021758) for instrument access, and assistance with sample and data analysis.

We acknowledge the supply of 25 adult urine specimens by FIND (Foundation for Innovative New Diagnostics; Geneva, Switzerland). We acknowledge the receipt of A194-01 antibody from Dr. Abraham Pinter at Rutgers New Jersey Medical School.

## Author Contributions

**Conceptualization:** Anita G. Amin, Prithwiraj De, Delphi Chatterjee.

**Formal analysis:** Anita G. Amin, Prithwiraj De, Barbara Graham.

**Funding acquisition:** Delphi Chatterjee.

**Methodology:** Anita G. Amin, Prithwiraj De.

**Project administration:** Delphi Chatterjee.

**Resources:** Brooke L. Jensen, Emmanuel Moreau, Delphi Chatterjee.

**Supervision:** Delphi Chatterjee.

**Validation:** Anita G. Amin.

**Writing – original draft:** Anita G. Amin, Delphi Chatterjee.

**Writing – review & editing:** Prithwiraj De, Barbara Graham, Emmanuel Moreau.

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
