## [Decision Letter · Decision Letter 0]

1 Sep 2022

PONE-D-22-15293Overcome Low Levels of Detection Limit and Choice of Antibody Affects Detection of Lipoarabinomannan in Pediatric TuberculosisPLOS ONE

Dear Dr. Chatterjee,

Thank you for submitting your manuscript to PLOS ONE. After careful consideration, we feel that it has merit but does not fully meet PLOS ONE’s publication criteria as it currently stands. Therefore, we invite you to submit a revised version of the manuscript that addresses the points raised during the review process.

We look forward to receiving your revised manuscript.

Kind regards,

Mao-Shui Wang

Academic Editor

PLOS ONE

Journal Requirements:

“This work was funded through NIH AI R01 AI132680 (to DC) and NIH/NAID U19 AI109755 (to Dr. Molly Franke for providing us with the samples used for analysis).”

 “NO The funders had no role in study design, data collection and analysis, decision to publish, or preparation of the manuscript.”

Reviewers' comments:

Reviewer's Responses to Questions

**Comments to the Author**

1. Is the manuscript technically sound, and do the data support the conclusions?

Reviewer #1: Yes

Reviewer #2: Yes

2. Has the statistical analysis been performed appropriately and rigorously? 

Reviewer #1: Yes

Reviewer #2: Yes

3. Have the authors made all data underlying the findings in their manuscript fully available?

Reviewer #1: Yes

Reviewer #2: Yes

4. Is the manuscript presented in an intelligible fashion and written in standard English?

Reviewer #1: Yes

Reviewer #2: Yes

5. Review Comments to the Author

Reviewer #1: i review with interest the study titled "Overcome Low Levels of Detection Limit and Choice of Antibody Affects Detection of

Lipoarabinomannan in Pediatric Tuberculosis" on 91 urine samples from children aged 3-10 years. urine samples were analyzed for tuberculostearic acid (TBSA) by gas chromatography/mass spectrometry (GC/MS) and

capture ELISA (C-ELISA). the study is interesting and well written. but i have few comments that could improve the manuscript:

1- the introduction is too long and should be shortened.

2- write the type of the study?

3- what is the duration of the study?

4- what are the exclusion criteria?

5- what are the primary and secondary outcomes of the study?

6- how did you estimate the sample size? and what is the power of the study?

Reviewer #2: Challenges with specimen collection and bacteriological confirmation of TB in young children, due to the paucibacillary nature of TB disease in this age group and the lack of highly sensitive point-of care tests. There is no gold-standard test available for children, and many of the older testing methods are often time-consuming and inaccurate. Using urine samples is a non-invasive, non-sputum-based diagnostic assay and useful method of detecting TB in children. The study indicates that for pediatric TB diagnosis, an ELISA based assay (which uses larger sample volumes) may be more appropriate where sample concentration can be applied. All children samples in this study were smear negative make the positive results in urine sample is more valuable. The study populations were very clear and appropriately. TB group, and non TB group based on 5 combination parameters: clinical symptom, X-ray, exposure history, TST and bacteriological evidence (microscopy, culture). Properly designed positive and negative control samples. Urine positive control for LAM spiked with CDC1551 LAM. The same urine sample was used as a negative control (no spike) for the urine background to derive a cutoff for the analysis of clinical samples. The used antibodies (mouse mAb CS35 IgG3, mAb A194-01 IgG1) were well calibrated. Techniques (C-ELISA , GC/MS) applied according to standard procedures. The statistical analysis has been performed appropriately and rigorously (BJ76 was tested twice and the OD450 values averaged. Confidence intervals for sensitivity and specificity were calculated using Wilson’s method. Statistical analyses were conducted in R open source software version 4.1.1). In 2022, WHO strongly recommends using LF-LAM to assist in the diagnosis of TB disease in HIV-positive adults and children with signs and symptoms of TB (pulmonary and/or extrapulmonary). The result of study make a suggestion that the use of urine specimens may consider as a tool for definitive diagnosis of tuberculosis, not only as a diagnostic assist. Unfortunately, there was no HIV status of the pediatric patient in the study. If the author analyzes the results by age group, especially in the group of children under 5 years old, it may have an additional contribution in screening, diagnosis and treatment.

6. PLOS authors have the option to publish the peer review history of their article (what does this mean?). If published, this will include your full peer review and any attached files.

Reviewer #1: **Yes: **Doaa El Amrousy

Reviewer #2: No

---

## [Author Response · Author response to Decision Letter 0]

19 Sep 2022

To The Editor

PLOS One

Ref Manuscript [PONE-D-22-15293]

Thank you for a very thorough review of our manuscript in PLOS One. [PONE-D-22-15293]

We have now carefully edited, reduced the Introduction by 10% and answered to the reviewers concerns as best as possible. The responses are all hilited in red.

• An unmarked version of your revised paper without tracked changes. You should upload this as a separate file labeled 'Manuscript'..

We look forward to receiving your revised manuscript.

Journal Requirements:

“This work was funded through NIH AI R01 AI132680 (to DC) and the samples analyzed in this study were provided under NIH/NAID U19 AI109755 to Dr. Molly F Franke (Harvard Medical School).”

This statement has been removed from the acknowledgement section

 “NO The funders had no role in study design, data collection and analysis, decision to publish, or preparation of the manuscript.”

This statement in correct as stated, funding related statement has been removed from the Acknowledgement section of the Manuscript

Funding Information provided is correct.

The Supplementary Files (Supporting Information) provided with this manuscript contain all the data presented in this study. There is no other repository information for data availability.

Captions for supporting Information are revised and included at the end of the manuscript after the References and updated in-text citations to match 

Not Applicable, All References have been cross checked.

Reviewers' comments:

Reviewer's Responses to Questions

Comments to the Author

1. Is the manuscript technically sound, and do the data support the conclusions?

Reviewer #1: Yes

Reviewer #2: Yes

2. Has the statistical analysis been performed appropriately and rigorously?

Reviewer #1: Yes

Reviewer #2: Yes

3. Have the authors made all data underlying the findings in their manuscript fully available?

Reviewer #1: Yes

Reviewer #2: Yes

4. Is the manuscript presented in an intelligible fashion and written in standard English?

Reviewer #1: Yes

Reviewer #2: Yes

5. Review Comments to the Author

Reviewer #1: i review with interest the study titled "Overcome Low Levels of Detection Limit and Choice of Antibody Affects Detection of

Lipoarabinomannan in Pediatric Tuberculosis" on 91 urine samples from children aged 3-10 years. urine samples were analyzed for tuberculostearic acid (TBSA) by gas chromatography/mass spectrometry (GC/MS) and

capture ELISA (C-ELISA). the study is interesting and well written. but i have few comments that could improve the manuscript:

1- the introduction is too long and should be shortened. Edited and shortened

2- write the type of the study? Experimental Assay Assessment

3- what is the duration of the study? Samples were from the U19 (NIH/NAID U19 AI109755) program that originated in 2015,-Mesman, A.W., Soto, M., Coit, J. et al. Detection of Mycobacterium tuberculosis in pediatric stool samples using TruTip technology. BMC Infect Dis 19, 563 (2019). https://doi.org/10.1186/s12879-019-4188-8. Children recruited to participate in a pediatric TB Diagnostic Study between May 2015-Feb 2018 in Peru.

 91 urine sample cohort was sent to CSU in 2020, GC/MS was done in September 2021and C-ELISA completed in March 2022, manuscript incorporating all data was submitted in May 2022. Thus, the duration of experimental work described herein was 2020-2022.

4- what are the exclusion criteria? For our study-Not Applicable as the study design was applied by the U19 program which states. “Subjects recruited in the pediatric TB diagnostics study were less than 15 years of age with a history of contact with an adult with TB within the previous two years, presenting with TB like symptoms i.e., persistent cough for two weeks or longer, unexplained weight loss and fever with fatigue for more than a week. 

5- what are the primary and secondary outcomes of the study? The primary outcome of our study is LAM is present albeit in low levels in children with TB symptoms as tested by clinicians irrespective of culture/smear status and amounts of LAM is quantifiable by GC/MS. Secondary outcome is, sensitive immunoassay needs to be developed with more selective Abs, or sample concentration need to be performed to detect low levels of urinary LAM among children

6- how did you estimate the sample size? and what is the power of the study? Sample size was not predetermined, we received and analyzed what was available as left over and sent to CSU for our experiments.

Reviewer #2: Challenges with specimen collection and bacteriological confirmation of TB in young children, due to the paucibacillary nature of TB disease in this age group and the lack of highly sensitive point-of care tests. There is no gold-standard test available for children, and many of the older testing methods are often time-consuming and inaccurate. Using urine samples is a non-invasive, non-sputum-based diagnostic assay and useful method of detecting TB in children. The study indicates that for pediatric TB diagnosis, an ELISA based assay (which uses larger sample volumes) may be more appropriate where sample concentration can be applied. All children samples in this study were smear negative make the positive results in urine sample is more valuable. The study populations were very clear and appropriately. TB group, and non TB group based on 5 combination parameters: clinical symptom, X-ray, exposure history, TST and bacteriological evidence (microscopy, culture). Properly designed positive and negative control samples. Urine positive control for LAM spiked with CDC1551 LAM. The same urine sample was used as a negative control (no spike) for the urine background to derive a cutoff for the analysis of clinical samples. The used antibodies (mouse mAb CS35 IgG3, mAb A194-01 IgG1) were well calibrated. Techniques (C-ELISA , GC/MS) applied according to standard procedures. The statistical analysis has been performed appropriately and rigorously (BJ76 was tested twice and the OD450 values averaged. Confidence intervals for sensitivity and specificity were calculated using Wilson’s method. Statistical analyses were conducted in R open source software version 4.1.1). In 2022, WHO strongly recommends using LF-LAM to assist in the diagnosis of TB disease in HIV-positive adults and children with signs and symptoms of TB (pulmonary and/or extrapulmonary). The result of study make a suggestion that the use of urine specimens may consider as a tool for definitive diagnosis of tuberculosis, not only as a diagnostic assist. Unfortunately, there was no HIV status of the pediatric patient in the study. If the author analyzes the results by age group, especially in the group of children under 5 years old, it may have an additional contribution in screening, diagnosis and treatment. We agree with the reviewer, the shortfall of our study is that we had no children sample with HIV coinfection for comparison. When we initiated this work, our goal was to thoroughly investigate LAM distribution among HIV negative TB positive patients and we concentrated in samples originating in Peru where HIV incidence is >1%. The children cohort also originated in Peru, therefore, no HIV, we are now looking into a new pediatric sample source which will have HIV/TB coinfection. As suggested, we analyzed the data based on <5yrs>, and our analysis shows that LAM amounts are equally distributed among all ages. (see forest plot in Supplementary Fig S14). This also, fits our hypothesis that when low levels (+/-2-4ng) of LAM are detected, it is difficult to correlate with any factors such as age/smear and perhaps sex.

6. PLOS authors have the option to publish the peer review history of their article (what does this mean?). If published, this will include your full peer review and any attached files.

Do you want your identity to be public for this peer review? For information about this choice, including consent withdrawal, please see our Privacy Policy.

Reviewer #1: Yes: Doaa El Amrousy

Reviewer #2: No

---

## [Editor Report · Decision Letter 1]

26 Sep 2022

Overcome Low Levels of Detection Limit and Choice of Antibody Affects Detection of Lipoarabinomannan in Pediatric Tuberculosis

PONE-D-22-15293R1

Dear Dr. Chatterjee,

We’re pleased to inform you that your manuscript has been judged scientifically suitable for publication and will be formally accepted for publication once it meets all outstanding technical requirements.

Kind regards,

Mao-Shui Wang

Academic Editor

PLOS ONE
---

## [Editor Report · Acceptance letter]

30 Sep 2022

PONE-D-22-15293R1 

Overcome Low Levels of Detection Limit and
Choice of Antibody Affects Detection of Lipoarabinomannan in Pediatric Tuberculosis 

Dear Dr. Chatterjee:

I'm pleased to inform you that your manuscript has been deemed suitable for publication in PLOS ONE. Congratulations! Your manuscript is now with our production department. 

Kind regards, 

on behalf of

Dr. Mao-Shui Wang 

Academic Editor

PLOS ONE